# Fangchinoline Inhibits African Swine Fever Virus Replication by Suppressing the AKT/mTOR/NF-κB Signaling Pathway in Porcine Alveolar Macrophages

**DOI:** 10.3390/ijms25137178

**Published:** 2024-06-29

**Authors:** Guanming Su, Xiaoqun Yang, Qisheng Lin, Guoming Su, Jinyi Liu, Li Huang, Weisan Chen, Wenkang Wei, Jianxin Chen

**Affiliations:** 1Guangdong Provincial Key Laboratory of Veterinary Pharmaceutics Development and Safety Evaluation, Guangzhou 510642, China; su18344309327@163.com (G.S.); yangxiaoqun970414@163.com (X.Y.); lqs@stu.scau.edu.cn (Q.L.); 18960701297@163.com (G.S.); jinyi1202@163.com (J.L.); 2College of Veterinary Medicine, South China Agricultural University, Guangzhou 510642, China; 3State Key Laboratory of Veterinary Biotechnology, Harbin Veterinary Research Institute, Chinese Academy of Agricultural Sciences, Harbin 150069, China; highlight0315@163.com; 4Department of Biochemistry and Genetics, La Trobe Institute for Molecular Science, La Trobe University, Melbourne, VIC 3086, Australia; weisan.chen@latrobe.edu.au; 5State Key Laboratory of Swine and Poultry Breeding Industry, Agro-Biological Gene Research Center of Guangdong Academy of Agricultural Sciences, Guangzhou 510642, China

**Keywords:** African swine fever virus (ASFV), fangchinoline (FAN), inhibition, AKT, mTOR, NF-κB

## Abstract

African swine fever (ASF), caused by the African swine fever virus (ASFV), is one of the most important infectious diseases that cause high morbidity and mortality in pigs and substantial economic losses to the pork industry of affected countries due to the lack of effective vaccines. The need to develop alternative robust antiviral countermeasures, especially anti-ASFV agents, is of the utmost urgency. This study shows that fangchinoline (FAN), a bisbenzylisoquinoline alkaloid found in the roots of *Stephania tetrandra* of the family Menispermaceae, significantly inhibits ASFV replication in porcine alveolar macrophages (PAMs) at micromolar concentrations (IC_50_ = 1.66 µM). Mechanistically, the infection of ASFV triggers the AKT/mTOR/NF-κB signaling pathway. FAN significantly inhibits ASFV-induced activation of such pathways, thereby suppressing viral replication. Such a mechanism was confirmed using an AKT inhibitor MK2206 as it inhibited AKT phosphorylation and ASFV replication in PAMs. Altogether, the results suggest that the AKT/mTOR pathway could potentially serve as a treatment strategy for combating ASFV infection and that FAN could potentially emerge as an effective novel antiviral agent against ASFV infections and deserves further in vivo antiviral evaluations.

## 1. Introduction

African swine fever (ASF), a devastating and contagious infectious disease caused by the African swine fever virus (ASFV), poses a major threat to the sustainability of the domestic swine industry. ASF, initially endemic to sub-Saharan Africa, has now spread to numerous countries across the globe, particularly in Europe and Asia, significantly impacting the worldwide swine industry and resulting in substantial economic losses [1]. Due to continuous ASFV recombination and evolution, type Ⅱ strains, originally with high virulence have evolved into less virulent strains that are characterized by comparable transmissibility but reduced mortality. Their infection manifests as non-fatal, subacute, or chronic persistent diseases [2]. On the other hand, recombinant strains between type Ⅰ and type Ⅱ high virulence strains have led to more virulent strains [3], which cause more deadly diseases. Both naturally evolved less or more lethal strains pose significant challenges to timely diagnosis and effective control of ASF [4].

ASFVs mainly replicate within porcine alveolar macrophages (PAMs). Viral replication relies on a highly coordinated process, including controlled viral gene expression and complex interplay between host cellular pathways and viral proteins. The inflammatory response induced by the ASFV plays an important role in its pathogenesis. The ASFV has been reported to activate the cellular NF-κB pathway to stimulate macrophages to secrete TNFα, IL-1β, and other cytokines. The excessive pro-inflammatory cytokine production (called a “cytokine storm”) accounts for the most deaths among ASFV-infected pigs [5]. However, the molecular mechanisms underlying ASFV pathogenesis remain largely unknown.

To effectively control the ASFV threat, developing a safe and effective vaccine remains the optimal goal, which has been elusive for over a century, largely due to the insufficient understanding of ASFV pathogenesis and its interaction with host immune regulation. As a result, biosecurity measures and mass-culling policies continue to stand as the primary strategies for preventing and controlling ASF [6]. However, the global implementation of these methods can be quite challenging due to high economic costs and demanding technical requirements. The immature state of ASF vaccines and the difficulty in rapidly developing safe and effective vaccines underscore the urgent need for anti-ASFV drugs. Despite that, some natural compounds have been identified with inhibitory activities against ASFV infections in vitro, including, luteolin [7], dihydromyricetin [8], aloe-emodin [9], and toosendanin [10]; so far there are no commercially available drugs for combating ASFV infections clinically.

Protein kinase B (AKT), a downstream protein of phosphatidylinositol 3–kinase (PI3K), is a serine/threonine kinase that serves as a central node, modulating the functions of numerous downstream proteins associated with cellular survival, proliferation, migration, metabolism, and angiogenesis [11]. Many viruses, such as the hepatitis C virus (HCV) and porcine reproductive and respiratory syndrome virus (PRRSV), modulate the PI3K/AKT pathway to benefit their growth [12,13]. The mammalian target of rapamycin (mTOR) and nuclear factor kappa-B (NF-κB) are downstream effectors within the PI3K/AKT signaling pathway. The mTOR and NF-κB pathways seem closely related to viral infection [14,15]. The synthesis of the Newcastle disease virus (NDV) protein requires the activation of the PI3K/AKT/mTOR signaling pathway. [16]. The entry of Bovine herpesvirus-1 (BoHV-1) into the Madin-Darby bovine kidney cells is dependent on the PI3K/AKT/NF-κB signaling pathway [17]. Furthermore, severe lung injury induced by the influenza A (H1N1) pdm09 virus infection necessitates the activation of the mTOR/NF-κB/NLRP3/interleukin-1β (IL-1β) axis [18]. However, further research is necessary to clarify the functional relationship between AKT, mTOR, and NF-κB in the context of ASFV infections.

Fangchinoline (FAN, the chemical structure depicted in Figure 1A) is a bisbenzylisoquinoline alkaloid derived from the root of *Stephaniae tetrandrine* S. Moore (family-Menispermaceae) [19], which is known as Fangji in China, a traditional Chinese medicine with long historical use. This medicinal plant has been traditionally used for various ailments, including rheumatism, arthralgia, edema, beriberi, unfavorable urination, and eczema [20]. FAN and its derivatives were demonstrated to have fungicidal activity, positioning FAN as a potential lead compound for antifungal pesticides [21]. FAN exhibits anti-cancer effects, with mechanisms of action encompassing the inhibition of tumor cell proliferation and metastasis, as well as the induction of cell apoptosis and autophagy [19]. In recent years, FAN was found to possess antiviral activities against the porcine epidemic diarrhea virus (PEDV) [22], human immunodeficiency virus type 1 (HIV) [23], Middle East respiratory syndrome coronavirus (MERS-CoV), and severe acute respiratory syndrome coronavirus 2 (SARS-CoV-2) [24]. Nevertheless, the molecular mechanisms underlying FAN’s antiviral action remain largely unelucidated.

Our current investigation demonstrates a robust inhibitory effect of FAN on ASFV replication at micromolar concentrations in PAMs. Moreover, we show that FAN inhibits the ASFV infection by suppressing the AKT/mTOR/NF-κB signaling pathway, which is required by effective ASFV replication. This represents the first report on the mechanisms underlying FAN’s antiviral activities, providing valuable insights into the potential application of FAN as a novel antiviral agent.

## 2. Results

### 2.1. Fangchinoline (FAN) Inhibited ASFV Replication in PAMs

Using an MTT assay, the cytotoxicity of FAN was evaluated in PAMs. FAN did not impair PAM viability at concentrations ≤ 10 μM after 48 h treatment (Figure 1B). Based on the above findings, 10 μM of FAN was chosen as the highest concentration for subsequent experimental investigations. The CC_50_ value (The value of 50% cytotoxic concentration) of FAN was determined to be 34.6 μM on PAMs. Subsequently, the antiviral activity of FAN against the ASFV infection was assessed in PAMs using IFA at 48 hpi. FAN effectively suppressed the ASFV infection in a dose-dependent manner (Figure 1C,D). By counting infected cells in IFA images, the 50% inhibitory concentration (IC_50_) of FAN against the ASFV infections was determined to be 1.66 μM, with a corresponding selectivity index (SI) of 20.8. Enrofloxacin (ENRO), and a reported inhibitor of ASFV replication served as a positive antiviral drug control [25]. In our results, 70 μM of ENRO exhibited equal inhibition on ASFV replication to that of 3.3 μM FAN. The obtained results demonstrated that FAN possesses robust inhibitory properties against ASFV infections.

We further investigated the effects of FAN on ASFV replication by infecting PAMs with different virus MOI. Figure 2A,B demonstrate that treatment with FAN significantly reduced the increased viral B646L genomic copy number and early gene CP204L mRNA levels at three different ASFV infection doses. Subsequently, we examined the impact of FAN on the viral protein expression, viral genomic copy number, virus titer, and mRNA levels at 24 hpi and 48 hpi. In the infected control (ASFV), the ASFV-p30 protein expression levels, the increased viral B646L genomic copy number, and the early gene CP204L mRNA levels and the virus titer markedly increased from 24 to 48 hpi. The addition of FAN (1.1, 3.3, and 10 µM) significantly reduced the ASFV-p30 protein expression, viral B646L genomic copy number, CP204L mRNA levels, and virus titer at both 24 and 48 hpi, as shown in Figure 2C–F. Taken together, these findings demonstrate that FAN significantly suppresses ASFV replication in PAMs.

### 2.2. FAN Does Not Directly Inactivate ASFV

To formally exclude direct interaction between FAN and ASFV, a virucidal assay was conducted, in which FAN at 100 µM was pre-incubated with an ASFV stock (1 × 10^5^ TCID_50_) for 1 h at 37 °C, and then the mixture was 1000-fold diluted with fresh medium before incubating with PAMs for 48 h (Figure 3A). To eliminate the possibility of antiviral activity at a low concentration of FAN, PAMs were incubated in the mixture with 0.1 μM FAN and 1 × 10^2^ TCID_50_ ASFV for 48 h at 37 °C, as a control. Our results showed that 0.1 μM of FAN showed no detectable inhibitory effect on ASFV replication. Meanwhile, ASFV stock treated with 100 µM FAN did not affect virus replication in PAMs, suggesting that FAN did not directly inactivate ASFV virions (Figure 3B,C).

### 2.3. FAN Pretreatment Significantly Reduces ASFV Replication

As FAN did not exhibit direct inactivation on ASFV virions, we wonder whether FAN could modulate the susceptibility of PAMs to ASFV infection. For this aim, PAMs were exposed to different concentrations of FAN for 3 h before ASFV infection (Figure 4A). The ASFV-p30 expression levels, viral B646L genomic copy number, and virus titer were significantly reduced by FAN pretreatment (Figure 4B,C). These findings suggest that FAN interacts with cellular components, thereby reducing the replication of ASFV.

### 2.4. FAN Inhibits ASFV Replication by Suppressing the AKT/mTOR Signaling Pathway

As FAN exhibited inhibition on ASFV replication in the pretreatment mode and did not interact with ASFV virions, we, therefore, wondered whether FAN acts on a certain cellular signaling pathway to inhibit ASFV infection. FAN has been reported to attenuate tumor migration and invasion by downregulating the c-Met/PI3K/AKT/mTOR and Wnt/β-catenin signaling pathways [26]. Inhibition of AKT phosphorylation has been demonstrated to exert anti-ASFV activity [27]. These reports prompted us to examine the relationship between the AKT/mTOR signaling pathway and FAN’s antiviral activity. We found that FAN inhibited cellular p-AKT and p-mTOR levels in a dose-dependent manner in PAMs (Figure 5A), indicating that FAN could suppress the cellular AKT/mTOR signaling pathway. Notably, the p-AKT and p-mTOR levels were significantly increased at 24 hpi, and FAN treatment significantly decreased the ASFV-induced p-AKT and p-mTOR levels (Figure 5B), suggesting that ASFV infection upregulates the AKT/mTOR signaling pathway which is suppressed by FAN. To confirm this finding, we subsequently evaluated the anti-ASFV activity of AKT inhibitor MK2206. Indeed, MK2206 (1 μM) significantly attenuated p-AKT, p-mTOR, and ASFV-p30 protein expression in both mock-infected and ASFV-infected PAMs, indicating that AKT is upstream of mTOR (Figure 5C,D). Of particular note, mTORC1 inhibitor rapamycin treatment reduced the p-mTOR expression in PAMs and also reduced the p-mTOR and ASFV-p30 protein expression in ASFV-infected PAMs (Figure 5E,F). These results showed that inhibition of p-AKT or p-mTOR levels results in suppressed ASFV replication. Taken together, our findings indicate that FAN inhibits ASFV replication by suppressing the AKT/mTOR signaling pathway.

### 2.5. FAN Inhibits ASFV Replication by Suppressing ASFV-Induced AKT/mTOR/NF-κB Signaling Pathway

Reports indicate that the ASFV infection has the ability to activate the NF-κB signaling pathway [28], and the NF-κB inhibitor BAY11-7082 effectively suppressed the replication of ASFV as well as the expressions of IL-1β and interleukin-8 (IL-8) [29]. Meanwhile, FAN effectively suppresses NF-κB activation induced by tumor necrosis factor-α (TNF-α) through attenuating phosphorylated IκB kinase (IKK) and p65 in human chronic myeloid leukemia cells (KBM5 cells) [30]. We hypothesized that the anti-ASFV activity exhibited by FAN is intimately associated with the NF-κB signaling pathway. Our findings demonstrated that the ASFV infection indeed enhanced the activity of the NF-κB signaling pathway, reflected by increased p-p65 and p-IκBα expression (Figure 6B). FAN inhibited cellular p-IκBα and p-p65 expression in a dose-dependent manner in both mock-infected and ASFV-infected PAMs (Figure 6A,B). Particularly, mTORC1 inhibitor RAPA at 5 µM also inhibited cellular expressions of p-p65 and p-IκBα in both mock-infected and ASFV-infected PAMs (Figure 6C,D). Meanwhile, the NF-κB inhibitor BAY11-7082 (BAY) at 5 µM inhibited cellular expressions of p-p65, p-IκBα, and ASFV replication in PAMs, reflected by decreased viral p30 protein expression (Figure 6E,F). These findings suggest that during ASFV infection, the NF-κB signaling pathway is likely downstream of the mTOR, and FAN can inhibit the mTOR/ NF-κB signaling pathway induced by ASFV infection. In addition, we explored the effects of the ASFV infection and FAN treatment on the expression profiles of cytokines (IL-1β and TNF-α) regulated by the NF-κB in PAMs. Our data showed that ASFV infections significantly upregulated mRNA levels of IL-1β and TNF-α in PAMs, and FAN robustly inhibited the increase of IL-1β and TNF-α mRNA induced by the ASFV infection (Figure 6G,H). These findings suggest that the ASFV infection stimulates the mTOR/NF-κB signaling pathway, and FAN inhibits ASFV replication by suppressing the AKT/mTOR/NF-κB signaling pathway.

## 3. Discussion

African swine fever (ASF), as a notoriously contagious disease, is characterized by its high morbidity and mortality. ASF epidemics have resulted in substantial economic losses in the global swine industry. Currently, there are no commercially available vaccines and antiviral drugs, which poses a significant challenge against ASFV infections. The current research marks the first demonstration of FAN’s powerful inhibitory actions against ASFV replication with 1.66 µM of IC_50_ value in PAMs. Mechanistically, we demonstrate that the anti-ASFV activity exhibited by FAN is closely related to its suppression of the ASFV-activated AKT/mTOR/NF-κB signaling pathway.

FAN was reported to have anti-tumor activity by modulating multiple cellular signaling cascades, of which the AKT pathway is the best understood [31,32]. During the entry of ASFV into PAMs and Vero cells, activation of the PI3K/AKT pathway has been demonstrated. AKT is phosphorylated at 5 mpi (minute post-infection), reaching a maximum of 30 mpi [27]. Furthermore, treatment of PI3K/AKT inhibitor LY294002 led to a 45% decrease in ASFV uptake at 60 mpi, and a 95% decrease in infected cells at 6 hpi. These findings suggest that the activation of PI3K/AKT is involved in ASFV entry [27,33,34]. However, the connection between FAN’s antiviral activity against ASFV infections and the AKT pathway remained unclear. Our study showed that ASFV infections promoted AKT phosphorylation in PAMs, and FAN significantly inhibited AKT phosphorylation in both mock-infected and ASFV-infected PAMs. Meanwhile, the AKT inhibitor MK2206 also inhibited AKT phosphorylation and ASFV replication in PAMs (Figure 5). Our results suggest that FAN inhibits ASFV replication in PAMs by suppressing the AKT signaling pathway.

mTOR, a serine/threonine protein kinase, controls essential cellular processes, including protein synthesis, metabolism, and autophagy [35], and is the core component of two mTOR-related protein complexes: mTORC1 and mTORC2 [36]. It was reported that mTORC1, but not mTORC2, could be exploited by viruses to enhance viral protein expression [37,38]. AKT is one of the critical signaling molecules regulating mTORC1 activity, and some virus infections activate the AKT/mTOR pathway to enhance the viral protein expression, including JC polyomavirus (JCPyV), human cytomegalovirus (HCMV), and MERS-CoV [39,40,41]. Some reported inhibitors targeting the AKT/mTOR pathway significantly inhibited JCPyV and MERS-CoV replication in vitro [39,41]. However, for PRRSV infection, the activated AKT pathway promotes PRRSV replication, yet the activated mTOR pathway suppresses PRRSV replication in MARC-145 cells, which suggests that AKT and mTOR are involved in PRRSV replication independently [42]. During ASFV infection, the relationship between AKT and mTOR remains poorly understood. In our study, the ASFV infection increased mTOR phosphorylation, and the AKT inhibitor MK2206 suppressed mTOR phosphorylation in PAMs (Figure 5C,D). Meanwhile, mTORC1 inhibitor rapamycin inhibited ASFV replication while it suppressed mTOR phosphorylation in PAMs (Figure 5E,F). These findings suggest that the AKT/mTOR pathway might be a therapeutic target against ASFV infection, similar to the cases against MERS-CoV and JCPyV infections [39,41].

Various virus infections activate NF-κB, such as IAV and HIV. The NF-κB is a crucial transcription factor that plays a pivotal role in regulating inflammatory cytokine expression [43,44]. Recently, a transcriptome analysis using single-cell RNA-sequencing technology showed that the ASFV infection induces PAMs to produce inflammatory cytokines via activating the NF-κB pathway [45]. AKT phosphorylation may lead to mTOR and NF-κB activation, and an intricate interplay between AKT, mTOR, and NF-κB involves crosstalk and complex regulatory mechanisms [17,46,47]. However, the relationship between AKT/mTOR and NF-κB during ASFV infection remains poorly established. Our study showed that ASFV infections activated the NF-κB pathway. FAN suppressed the NF-κB pathway in PAMs (Figure 6). Meanwhile, mTORC1 inhibitor rapamycin suppressed the NF-κB pathway in PAMs (Figure 6C,D). The NF-κB inhibitor BAY11-7082 suppressed the replication of ASFV (Figure 6E,F). Notably, the decrease of p-p65 and p-IκBα expression in FAN-treated ASFV-infected PAMs might be a direct result of FAN’s inhibition on the NF-κB pathway, or an indirect result of its anti-ASFV effect, or both, as we observed that FAN was capable of inhibiting p-p65 and p-IκBα basal levels in PAMs (Figure 6). Collectively, our findings suggest that the AKT/mTOR/NF-κB pathway could be a potential therapeutic target against ASFV infection, and FAN holds promising therapeutic potential as it demonstrates the ability to effectively inhibit the AKT/mTOR/NF-κB signaling pathway. A schematic summary of the inhibitory mechanisms of FAN on ASFV replication is shown in Figure 7.

FAN, characterized by low toxicity and high LD_50_ (median lethal dose) values (LD_50_ > 50 mg/kg, intraperitoneal injection in mice), has been established as a safe compound [48,49]. It was reported that FAN at concentrations from 1 to 3 mg/kg could significantly attenuate the tumor volume and weight in mice by suppressing the PI3K/AKT/mTOR pathway [50], suggesting it is likely feasible to use FAN for in vivo application.

In spite of the robust inhibition of FAN on ASFV replication in PAMs, its in vivo antiviral potential is yet to be determined as that could be affected by its pharmacokinetic properties and its tolerated dose. Therefore, it will be necessary to conduct well-designed pre-clinical studies to investigate FAN’s antiviral effect against ASFV infections in swine.

## 4. Materials and Methods

### 4.1. Cell and Virus Strains

A total of four ASFV-negative 4-week-old piglets (weighing 7~8 kg) were purchased from Xinli Pig Farm (Wuzhou, China). Primary porcine alveolar macrophages (PAMs) were isolated from the lungs of these piglets, utilizing a previously established method [51]. In brief, PAMs were collected from the lungs using a pre-cooled RPMI-1640 medium (Gibco, UT, USA) as the lavage solution. PAM suspension was centrifuged, resuspended in a cell freezing medium (Dimethyl sulfoxide: Fetal bovine serum = 9:1), dispensed into cryopreservation tubes (1 × 10^7^ PAMs/tube), and preserved in liquid nitrogen. The Ⅱ ASFV (GZ201801 strain, GenBank accession number MT496893.1) utilized in this study was maintained at the Biosafety Level 3 (BSL-3) laboratory located at the South China Agricultural University in Guangzhou, China. The experimental procedures for this study, which involved the handling of live ASFV, were strictly conducted within the BSL-3 laboratory.

### 4.2. Antibodies, Chemicals, and Other Reagents

The anti-ASFV p30 antibody used in the Western blot was a generous gift from Prof. Changjiang Weng from the Harbin Veterinary Research Institute, Chinese Academy of Agricultural Sciences. The anti-mouse IgG antibody conjugated with Alexa Fluor^®^488 (green) was purchased from Cell Signaling Technology (Boston, MA, USA). Horseradish peroxidase-labeled goat anti-mouse IgG (A0216), horseradish peroxidase-labeled goat anti-rabbit IgG (A0208), an anti-α-tubulin mouse monoclonal antibody (AF0001), and a BeyoECL Plus Chemiluminescence kit (P0018M) were obtained from Beyotime Biotechnology (Shanghai, China). An anti-mTOR antibody (mTOR), anti-Phospho-mTOR (Ser2448) antibody (p-mTOR), anti-NF-κB p65 antibody (p65), anti-Phospho-NF-κB p65 (Ser536) antibody (p-p65), anti-IκBα antibody (IκBα), anti-Phospho-IκBα (Ser32) antibody (p-IκBα), anti-AKT antibody (AKT), and anti-Phospho-AKT (Ser473) antibody (p-AKT) were obtained from Cell Signaling Technology (Boston, MA, USA). The antibodies were stored at −20 °C, diluted using the primary Antibody Dilution Buffer (Beyotime, Haimen, China), and formulated at 1 μg/mL. The diluted antibodies were stored at 4 °C. Fangchinoline (FAN) was bought from Chengdu Pufei De Biotech Co., Ltd. (Chengdu, China), and MK2206, rapamycin, and BAY11-7082 were purchased from Med Chem Express (Shanghai, China). The chemical powders were stored at 4 °C. They were dissolved in DMSO at 10 mg/mL. The diluted chemicals were stored at 4 °C.

### 4.3. Cellrual Cytotoxicity Assay

The cytotoxicity of FAN on PAMs was evaluated through the application of an MTT assay. In brief, PAMs were incubated with different concentrations of FAN at 37 °C for 48 h. The PAMs were then incubated with the MTT (Sigma-Aldrich, Saint Louis, MO, USA) solution (0.5 mg/mL in PBS) at 37 °C for 4 h, followed by the addition of DMSO. The microplate reader (Thermo Fisher Scientific, Waltham, MA, USA) was applied to measure the OD value at 570 nm. The 50% cytotoxic concentration (CC_50_) was analyzed using a GraphPad Prism 8.0.

### 4.4. Quantitative Real-Time PCR (qPCR)

The extraction of total DNA and RNA from the cells or culture supernatants was performed using the FastPure Cell/Tissue DNA isolation Mini Kit (Vazyme, Nanjing, China) or the Total RNA Rapid Extraction Kit (Fastagen, Shanghai, China), respectively, in accordance with the manufacturer’s instructions [10]. In brief, the samples were collected in 2 mL microfuge tubes, lysis solution was added, and the lysates were carefully aspirated into the separation columns and centrifuged. Subsequently, elution buffer was added and centrifuged to elute the total DNA or RNA. The extracted RNA was converted into cDNA using a reverse transcription kit sourced from Genstar (Beijing, China). The resulting cDNAs, or DNA templates, were then amplified by employing the 2×RealStar Green Power Mixture sourced from Genstar in Beijing, China, on LightCycler96 manufactured by Roche in BSL, CH. The classical 2^−ΔΔCT^ method was used to calculate the relative mRNA expression [52]. The increased viral B646L genomic copy number was calculated using the standard curve. The following are the primer sequences: B646L (5′-GGAAATTCATTCACCAAATCCTT-3′ and 5′-CTTCGGCGAGCGCTTTATCAC-3′); ASFV-RT-CP204L (5′-TGCACATCCTCCTTTGAAACAT-3′ and 5′-TCTTTTGTGCAAGCATATACAGCTT-3′); GAPDH (5′-CCTTCCGTGTCCCTACTGCCAAC-3′ and 5′-GACGCCTGCTTCACCACCTTCT-3′) [10].

### 4.5. Indirect Immunofluorescence Assay (IFA)

PBS washed the treated PAMs twice and then fix them in 4% paraformaldehyde for 15 min, subsequently permeabilized in 0.25% Triton X-100 for 10 min. After permeabilizing, the cells were blocked with 1% bovine serum albumin (BSA) for 60 min at room time. The PAMs were further incubated with a mouse monoclonal antibody against the ASFV-p30 at 4 °C overnight, then incubated at RT for 1 h with a goat anti-mouse secondary antibody conjugated with Alexa Fluor^®^ 488 (green) at a 1: 1000 dilution. Nuclei were counterstained with DAPI (300 nM; Sigma-Aldrich, Saint Louis, MO, USA) (blue). The fluorescence microscope (Leica, Heidelberg, Germany) was applied to capture immunofluorescence. In IFA images, fluorescence spot amounts of the nucleus (blue), and ASFV-p30 (green) in infected PAMs were calculated using the image J software 1.8.0, respectively.

### 4.6. Western Blot Analysis

The PAMs were lysed in the RIPA lysis buffer (supplied by Beyotime, Haimen, China) at 4 °C. The lysate was then centrifuged at 15,000× *g* for 10 min. Finally, the protein content within the supernatant was precisely quantified using a BCA protein assay kit from Beyotime, Haimen, China. Following the manufacturer’s guidelines for the BCA protein assay kit, the protein concentrations of the samples were determined by referencing the standard curve [53]. The SDS loading buffer was added, and the mixture was boiled for 10 min. The samples were subjected to electrophoresis on 10% SDS-PAGE gels and subsequently transferred onto polyvinylidene-fluoride (PVDF) membranes (Millipore, Boston, MA, USA). Following blocking with 5% skimmed milk (Beyotime, Haimen, China), the membranes were incubated overnight with antibodies, washed extensively, and then incubated with HRP-conjugated goat anti-mouse or anti-rabbit IgG (H–L) secondary antibodies (1:5000). The protein bands present on the PVDF membranes were analyzed using the K 6000 mini chemiluminescence imager supplied by Beijing Ke Chuang Rui Xin, Beijing, China.

### 4.7. Antiviral Activity Assay

PAMs were inoculated with 0.1 MOI ASFV for 2 h. Supernatants were discarded, followed by substitution with RPMI-1640 media containing different concentrations of testing compounds. The samples were gathered at the specified time intervals in order to determine the viral genomic copy number, viral mRNA level, and viral titer through the application of qPCR and endpoint dilution assay, respectively. IFA was utilized to determine the percentage of ASFV-infected cells, while the Western blot was employed to assess the viral protein expression.

### 4.8. Statistical Analysis

All data presented in this study are expressed as mean ± standard deviation (SD). Statistical significance was evaluated through the application of Student’s *t*-test in comparisons involving two distinct groups. All statistical analyses were performed using the GraphPad Prism 8 software. * *p*< 0.05, ** *p* < 0.01, and *** *p* < 0.001 were considered as statistically significant differences.

## 5. Conclusions

In conclusion, our research reveals that FAN effectively inhibits ASFV replication in PAMs at micromolar concentrations by inhibiting the ASFV-activated AKT/mTOR/NF-κB signaling pathway. FAN emerges as a promising candidate for the treatment of ASFV infections, and inhibiting the AKT/mTOR pathway could represent a novel approach for developing effective anti-ASFV agents. Given that the in vivo antiviral effect of one compound is closely associated with a variety of factors, including its pharmacokinetic properties, effective concentration in virus-infected tissues, and its tolerated dose, it will be necessary to conduct well-designed pre-clinical studies to investigate FAN’s antiviral effect against ASFV infections in swine.

## Figures and Tables

**Figure 1 ijms-25-07178-f001:**
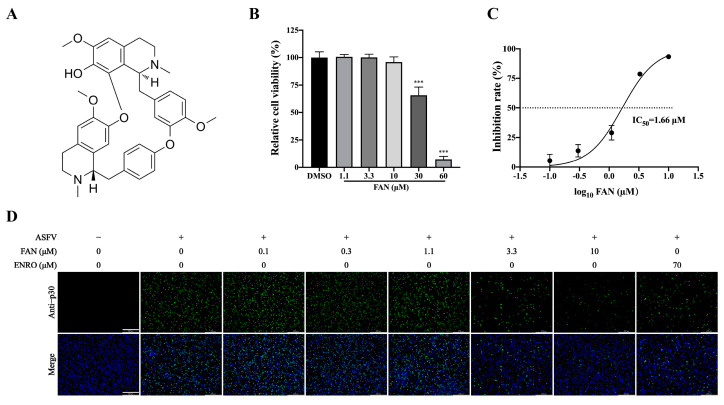
Cell cytotoxicity and anti−ASFV activity of fangchinoline (FAN). (**A**) Chemical structure of FAN. (**B**) PAMs were exposed to media containing different concentrations of FNA for 48 h, and the cellular viability was measured using an MTT assay. (**C**,**D**) PAMs were treated with media containing various concentrations of FAN for 48 h after ASFV infection (0.5 MOI), followed by IFA using the ASFV−p30−specific antibody. Scale bar: 250 μm. *** *p* < 0.001, compared with the corresponding viral control.

**Figure 2 ijms-25-07178-f002:**
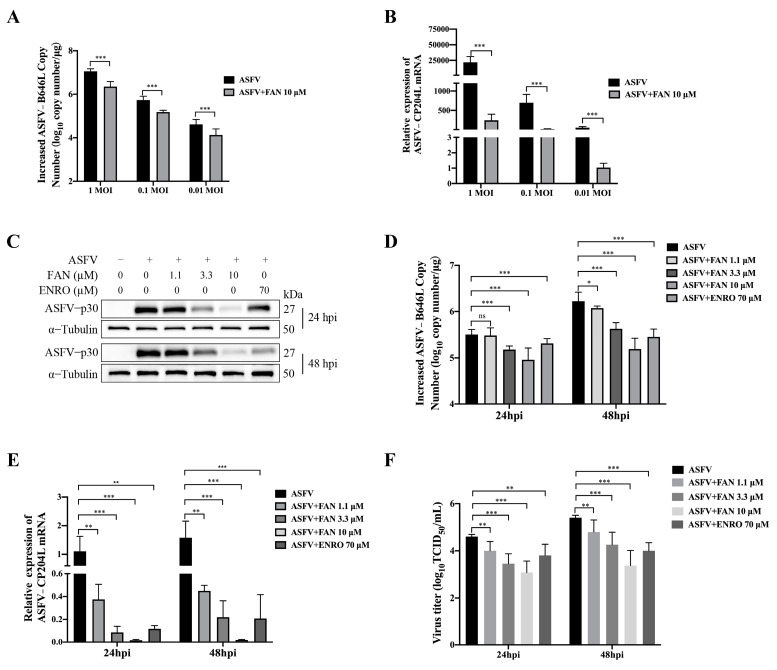
FAN inhibits ASFV replication in PAMs. (**A**,**B**) PAMs were treated with media containing 10 μM of FAN following ASFV infection (0.01, 0.1 or 1 MOI). At 24 hpi, the samples were gathered and subjected to qPCR analysis to assess increased viral B646L genomic copy number (**A**) and early gene CP204L mRNA levels (**B**). (**C**–**F**) PAMs were exposed to ASFV (0.1 MOI) for 2 h at 37 °C, followed by incubation in fresh media containing various concentrations of FAN. The samples were analyzed for ASFV−p30 protein expression, increased viral B646L genomic copy number, early gene CP204L mRNA level, and viral titer using Western blot, qPCR, and the end−point dilution assay, respectively, at indicated time points. * *p* < 0.05, ** *p* < 0.01, *** *p* < 0.001, ns means no significant difference, compared with the corresponding viral control.

**Figure 3 ijms-25-07178-f003:**
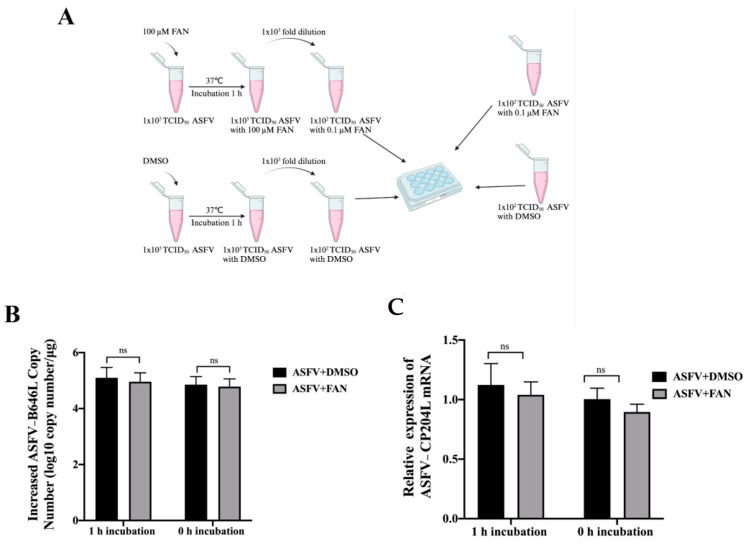
FAN does not directly interact with ASFV. The experimental design for FAN–ASFV interaction is shown in (**A**). The mixture containing 1 × 10^5^ TCID_50_ of ASFV and 100 μM FAN was incubated at 37 °C. Following an incubation period of 1 h, the mixture was diluted 1000-fold using fresh medium and subsequently used to infect PAMs for 2 h at 37 °C. As a control, PAMs were treated with 0.1 μM of FAN and 1 × 10^2^ TCID_50_ of ASFV for 2 h at 37 °C. The cells and supernatants were collected to detect increased viral B646L genomic copy number (**B**) and early gene CP204L mRNA level (**C**) at 48 hpi. ns means no significant difference, compared with the corresponding viral control.

**Figure 4 ijms-25-07178-f004:**
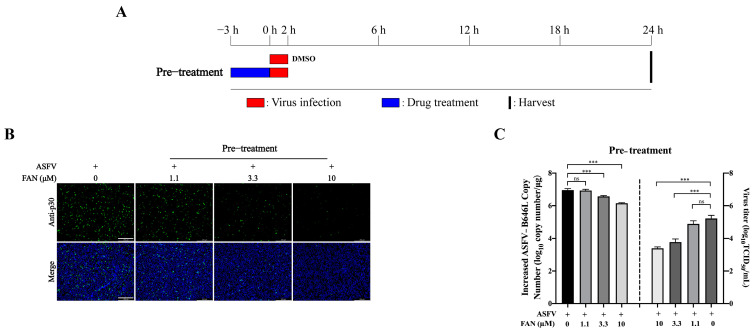
FAN pretreatment significantly reduces ASFV replication. (**A**) Scheme for pretreatment experiment. PAMs were subjected to media containing different concentrations of FAN for 3 h before ASFV infection at 0.1 MOI. At 24 hpi, the samples were collected for detection of ASFV−p30 expression (**B**), increased viral B646L genomic copy number, and virus titer (**C**). Scale bar: 250 μm. *** *p* < 0.001, ns means no significant difference, compared with the corresponding viral control.

**Figure 5 ijms-25-07178-f005:**
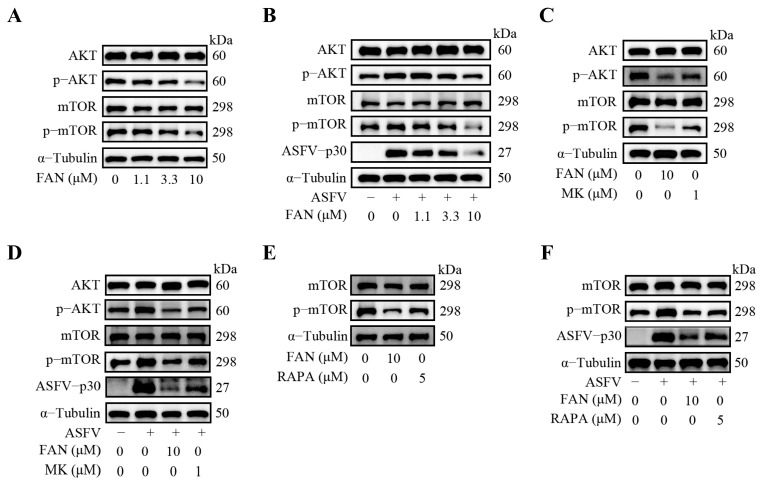
FAN inhibits ASFV replication by suppressing the AKT/mTOR signaling pathway. PAMs were treated with different concentrations of FAN (**A**) or 1 μM MK2206 (MK, a specific AKT inhibitor) (**C**) or 5 μM rapamycin (RARP, a mTORC1 inhibitor) (**E**) for 24 h, or infected with 0.1 MOI ASFV for 2 h, followed by treatment with FAN (**B**) or 1 μM MK2206 (**D**) or 5 μM rapamycin (**F**). The cells were collected at 24 h post compound treatment, and the indicated proteins were detected by Western blot.

**Figure 6 ijms-25-07178-f006:**
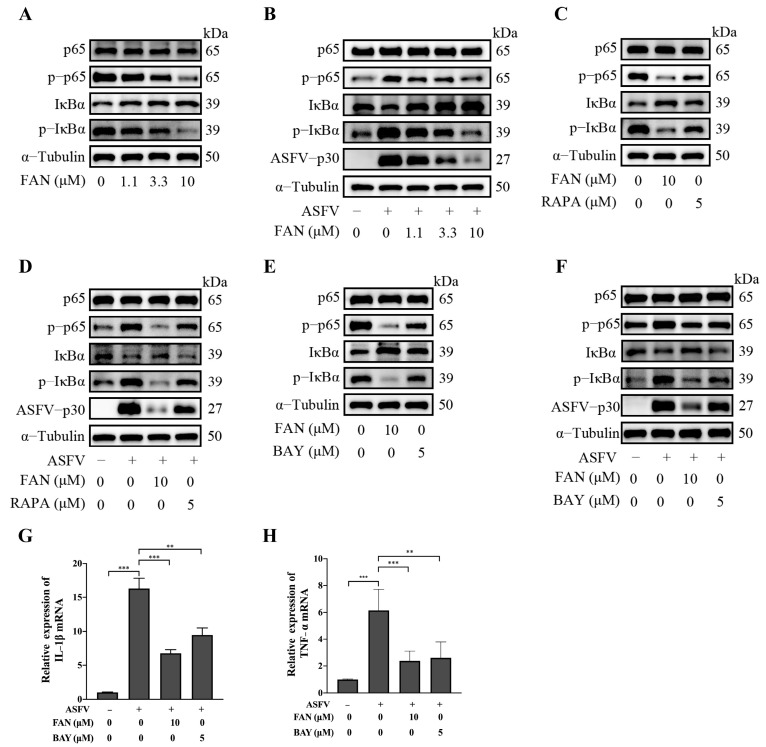
FAN inhibits ASFV replication by suppressing the ASFV−induced AKT/mTOR/NF−κB signaling pathway activation. PAMs were exposed to different concentrations of FAN (**A**) or 5 μM RAPA (**C**) or 5 μM, BAY11−7082 (BAY), a specific NF−κB inhibitor (**E**). PAMs were infected with the ASFV (0.1 MOI) for 2 h, followed by treatments with FAN (**B**), RARP (**D**), or BAY (**F**). After 24 h treatment with FAN, the cells were harvested to assess the expression levels of the indicated proteins by Western blot. The supernatants and cells were analyzed for IL−1β and TNF−α mRNA levels (**G**,**H**) ** *p* < 0.01, *** *p* < 0.001, compared with the corresponding viral control.

**Figure 7 ijms-25-07178-f007:**
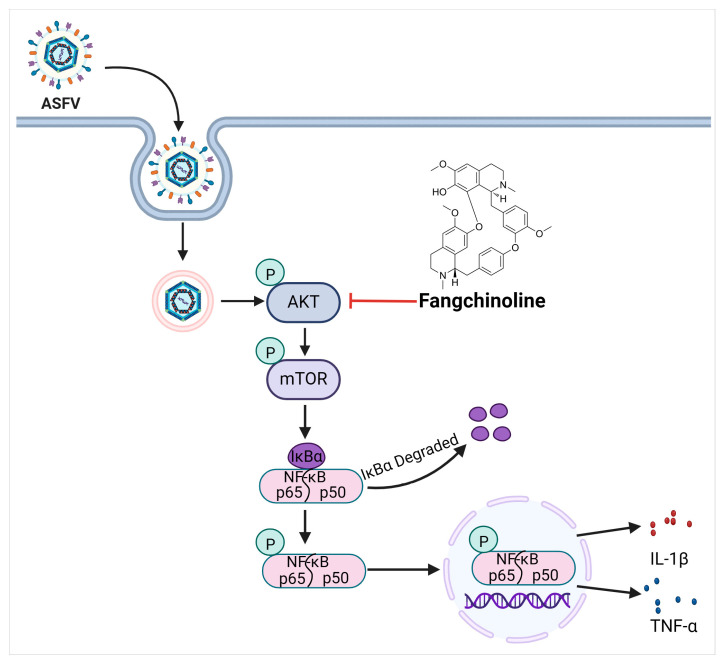
Schematic overview of the inhibitory mechanisms of FAN on the replication of ASFV. ASFV infection activates the cellular AKT/mTOR/NF-κB signaling pathway, and FAN suppresses AKT phosphorylation and thereby inhibits ASFV replication in PAMs.

## Data Availability

All data related to this study are included in the article. Other data can be obtained from the corresponding author if reasonably required.

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
