# Peer review of "Fangchinoline Inhibits African Swine Fever Virus Replication by Suppressing the AKT/mTOR/NF-κB Signaling Pathway in Porcine Alveolar Macrophages"

_ijms, 2024, doi:10.3390/ijms25137178_

Round 1

Reviewer 1 Report

Comments and Suggestions for Authors

IJMS-2976305 Peer Review Report
African swine fever remains a highly contagious disease with high morbidity and
mortality. Currently, this disease has no treatment, which calls for developing
strategies to mitigate its global importance. The findings of this study suggest that
fangchinoline might be a promising novel antiviral agent or lead compound against
the African swine fever virus and deserves further in vivo antiviral evaluations. The
paper is well-written, but I have made suggestions to improve it further.
Abstract:
a. The authors should state the study's objectives clearly. This is important as it
gives readers an understanding of what the research is set to achieve.
b. Briefly introduce ASF and its significance in the global pork industry. This will
help set the context for readers who may not be familiar with the disease.
c. Consider mentioning the current lack of effective treatment options for ASF,
highlighting the urgency and importance of developing anti-ASFV drugs.
d. The authors should provide adequate background data or evidence on the
present status of African Swine Fever Vaccine research, the challenges
associated with vaccine development, and the necessity for alternative
treatment approaches. Addressing these issues will help your readers
appreciate the study's objectives.
e. Complete the abstract with a clear statement recapping the consequences of the
outcomes or results and the potential of FAN as an anti-ASFV agent.
Introduction:
Line 38-41: This sentence requires a citation.
Line 44-46: This sentence, too, requires a citation.
Line 56: AKT and all other abbreviations should be defined at first mention.
Line 61-63: This sentence requires a citation.
Results
a. Lines 92-93: This sentence needs to be reworded to make it clear to your readers.
Suggestion: Figure 1B shows no apparent cytotoxicity for FAN concentrations
≤ 10 μM on PAMs after 48 h treatment.
b. Lines 106-108: This sentence should be paraphrased for better clarity. (A)
Chemical structure of FAN. (B) PAMs were treated with media containing
different concentrations of FNA for 48 h, and cell viability was measured using
an MTT assay.
c. Lines 135: Replace the word ‘with’ with ‘and’.
d. Line 156: Write ‘Figure’ in plural (Figures).
Discussion
a. The discussion ended rather abruptly. I suggest you state any limitations
confronted throughout the study that may have impacted the analysis and
interpretation of the findings generated from this study.
b. Briefly mention future research directions or next steps, such as in vivo
evaluations or clinical trials.
c. Line 240: Delete the hyphen joining best and understood.
Materials and Methods
a. Lines 308-309: Rephrase this sentence for clarity. Suggestion: The procedures
for this study involving live ASFV were conducted within the ABSL-3
laboratory at South China Agricultural University (Guangzhou, China).
b. Line 341: Delete the letter ‘s’ in ‘followings’. It becomes ‘following’.
c. Line 371: Delete the letter ‘s’ in afterwards. It becomes ‘afterwards’.
d. Line 379: Delete the letter ‘s’ in student’s. It becomes ‘student’.
e. The authors should offer more information on preserving PAMs in liquid
nitrogen, which may be useful for other researchers who wish to work in this
area.
f. Again, the authors should offer more information on the concentrations,
temperature requirements and light conditions of reagents utilised here.
g. There is a need to elaborate on the extraction or isolation procedures of DNA/
RNA.
h. Provide information on the normalization method used for quantifying protein
expression levels.
i. State the statistical software utilised in the data analysis.
Conclusion
Ideally, this section should outline the study's findings and suggest implications for
implementation. The conclusion provided in this paper is too short, and the following
are areas that could be improved upon for clarity:
a. Give definite results to confirm the assertion that FAN actually suppresses
ASFV replication at micromolar dilutions.
b. Proffer any likely clinical applications and/ or implications of using FAN as a
new approach to treat ASFV infection. Address any challenges that may occur
in rendering these results into clinical practice.
c. State any constraints of the study that may negatively impact the interpretation
of the results or the likely usage of FAN as an antiviral agent.
References
Consider presenting references for established methods and protocols mentioned in
the paper, particularly if they are not commonly published. This will assist readers in
replicating the experiments accurately.

Author Response

Responses to Reviewers′ Comments

Reviewer 1

Reviewer #1:

African swine fever remains a highly contagious disease with high morbidity and mortality. Currently, this disease has no treatment, which calls for developing strategies to mitigate its global importance. The findings of this study suggest that fangchinoline might be a promising novel antiviral agent or lead compound against the African swine fever virus and deserves further in vivo antiviral evaluations. The paper is well-written, but I have made suggestions to improve it further.

Abstract:

  1. The authors should state the study's objectives clearly. This is important as it gives readers an understanding of what the research is set to achieve.
  2. Briefly introduce ASF and its significance in the global pork industry. This will help set the context for readers who may not be familiar with the disease.
  3. Consider mentioning the current lack of effective treatment options for ASF, highlighting the urgency and importance of developing anti-ASFV drugs.
  4. The authors should provide adequate background data or evidence on the present status of African Swine Fever Vaccine research, the challenges associated with vaccine development, and the necessity for alternative treatment approaches. Addressing these issues will help your readers appreciate the study's objectives.
  5. Complete the abstract with a clear statement recapping the consequences of the outcomes or results and the potential of FAN as an anti-ASFV agent.

As the reviewer suggested, we have now modified the Abstract as following:

African swine fever (ASF), caused by the African swine fever virus (ASFV), is one of the most important infectious diseases that causes high morbidity and mortality in pigs and substantial economic losses to the pork industry of affected countries due to the lack of effective vaccines. The need to develop alternative robust antiviral countermeasures, especially anti-ASFV agents, is of utmost urgency. This study shows that fangchinoline (FAN), a bisbenzylisoquinoline alkaloid found in the roots of Stephania tetrandra of the family Menispermaceae, significantly inhibits ASFV replication in porcine alveolar macrophages (PAMs) at micromolar concentrations (IC50=1.66 µM). Mechanistically, ASFV infection promotes the activation of the AKT/mTOR/NF-κB signaling pathway, and FAN significantly inhibits ASFV-induced activation of such pathway, thereby suppressing viral replication. Such a mechanism was confirmed using AKT inhibitor MK2206 as it inhibited AKT phosphorylation and ASFV replication in PAMs. Altogether, our findings suggest that the AKT/mTOR pathway could be a therapeutic target against ASFV infection and that FAN might be a promising novel antiviral agent against ASFV infections and deserves further in vivo antiviral evaluations.

Introduction:

Line 38-41: This sentence requires a citation.

As suggested, we have now added a citation in line 47, page 2, listed as No.3 in the References section: [3] Sun, E.; Zhang, Z.; Wang, Z.; He, X.; Zhang, X.; Wang, L.; Wang, W.; Huang, L.; Xi, F.; Huangfu, H.; Tsegay, G.; Huo, H.; Sun, J.; Tian, Z.; Xia, W.; Yu, X.; Li, F.; Liu, R.; Guan, Y.; Zhao, D.; Bu, Z., Emergence and prevalence of naturally occurring lower virulent African swine fever viruses in domestic pigs in China in 2020. Sci China Life Sci 2021, 64, (5), 752-765, http://dx.doi.org/10.1007/s11427-021-1904-4.

Line 44-46: This sentence, too, requires a citation.

As suggested, we have now cited a reference in line 49, page 2, listed as No.5 in the References section: [5] Hu, Z.; Tian, X.; Lai, R.; Wang, X.; Li, X., Current detection methods of African swine fever virus. Front Vet Sci 2023, 10, 1289676, http://dx.doi.org/10.3389/fvets.2023.1289676.

Line 56: AKT and all other abbreviations should be defined at first mention.

As suggested, we have now defined all abbreviations at their first mention, including AKT as “Protein kinase B”, mTOR as “mammalian target of rapamycin”, NF-κB as “nuclear factor kappa-B”, IL-1β as “interleukin-1β” (page 2, lines 72-85); and SARS-CoV-2 as “severe acute respiratory syndrome coronavirus 2” (line 99, page 2), CC50 value as “the concentration required to reduce normal cell viability by 50%” (lines 113-114, page 3), IL-8 as “interleukin-8” (line 218, page 6), TNF-α as “tumor necrosis factor-α” (line 218, page 6), LD50 as “median lethal dose” (line 311, page 9).

Line 61-63: This sentence requires a citation.

As suggested, we have now cited two references in line 80, on page 2, listed as No.14 an 15 in the References section: [14] Glitscher, M.; Benz, N. I.; Sabino, C.; Murra, R. O.; Hein, S.; Zahn, T.; Mhedhbi, I.; Stefanova, D.; Bender, D.; Werner, S.; Hildt, E., Inhibition of Pim kinases triggers a broad antiviral activity by affecting innate immunity and via the PI3K-Akt-mTOR axis the endolysosomal system. Antiviral Res 2024, 226, 105891, http://dx.doi.org/10.1016/j.antiviral. 2024.105891, and [15] Wu, C. M.; Mao, J. W.; Zhu, J. Z.; Xie, C. C.; Yao, J. Y.; Yang, X. Q.; Xiang, M.; He, Y. F.; Tong, X.; Litifu, D.; Xiong, X. Y.; Cheng, M. N.; Zhu, F. H.; He, S. J.; Lin, Z. M.; Zuo, J. P., DZ2002 alleviates corneal angiogenesis and inflammation in rodent models of dry eye disease via regulating STAT3-PI3K-Akt-NF-kappaB pathway. Acta Pharmacol Sin 2024, 45, (1), 166-179, http://dx.doi.org/10.1038/s41401-023-01146-y.

Results

  1. Lines 92-93: This sentence needs to be reworded to make it clear to your readers. Suggestion: Figure 1B shows no apparent cytotoxicity for FAN concentrations ≤ 10 μM on PAMs after 48 h treatment.

As suggested, we have now reworded this sentence as: FAN did not show apparent cytotoxicity at concentrations ≤ 10 μM on PAMs after 48 h treatment (lines 111-112, page 3).

  1. b. Lines 106-108: This sentence should be paraphrased for better clarity. (A) Chemical structure of FAN. (B) PAMs were treated with media containing different concentrations of FNA for 48 h, and cell viability was measured using an MTT assay.

As suggested, we have now modified the sentences as: PAMs were treated with media containing different concentrations of FNA for 48 h, and the cellular viability was measured using an MTT assay (lines 126-127, page 3).

  1. Lines 135: Replace the word ‘with’ with ‘and’.

Modified.

  1. Line 156: Write ‘Figure’ in plural (Figures).

Modified.

Discussion

  1. The discussion ended rather abruptly. I suggest you state any limitations confronted throughout the study that may have impacted the analysis and interpretation of the findings generated from this study.
  2. Briefly mention future research directions or next steps, such as in vivo evaluations or clinical trials.

As suggested, we have now added several lines to enrich the discussion in the end as following:

In spite of robust inhibition of FAN on ASFV replication in PAMs, its in vivo antiviral potential is yet to be determined as that could be affected by its pharmacokinetic properties, and its tolerated dose and half-life. Thereby, it will be necessary to conduct well designed pre-clinical studies to investigate FAN’s antiviral effect against ASFV infection in swine (lines 316-319, page 9).

  1. Line 240: Delete the hyphen joining best and understood.

Modified.

Materials and Methods

  1. Lines 308-309: Rephrase this sentence for clarity. Suggestion: The procedures for this study involving live ASFV were conducted within the ABSL-3 laboratory at South China Agricultural University (Guangzhou, China).

We have now rephrased this sentence as suggested (lines 337~338, page 10).

  1. Line 341: Delete the letter ‘s’ in ‘followings’. It becomes ‘following’.

Modified.

  1. Line 371: Delete the letter ‘s’ in afterwards. It becomes ‘afterwards’.

Modified.

  1. Line 379: Delete the letter ‘s’ in student’s. It becomes ‘student’.

Modified.

  1. The authors should offer more information on preserving PAMs in liquid nitrogen, which may be useful for other researchers who wish to work in this area.

As suggested, we have now offered more information on preserving PAMs in liquid nitrogen as the following:

PAM suspension was centrifugated, resuspended in cell freezing medium (Dimethyl sulfoxide: Fetal bovine serum=9:1), dispensed into cryopreservation tubes (1×107 PAMs/tube), and preserved in liquid nitrogen. (lines 332-334, page 9).

  1. Again, the authors should offer more information on the concentrations, temperature requirements and light conditions of reagents utilised here.

As suggested, we have now provided more information for the utilized reagents as following:

“Anti-mTOR antibody (mTOR), anti-Phospho-mTOR (Ser2448) antibody (p-mTOR), anti-NF-κB p65 antibody (p65), anti-Phospho-NF-κB p65 (Ser536) antibody (p-p65), anti-IκBα antibody (IκBα), anti-Phospho-IκBα (Ser32) antibody (p-IκBα), anti-AKT antibody (AKT) and anti-Phospho-AKT (Ser473) antibody (p-AKT) were purchased from Cell Signaling Technology, Inc (MA, USA). The antibodies were stored at -20℃, diluted by the primary Antibody Dilution Buffer (Beyotime, Haimen, China), and formulated at 1 μg/mL. The diluted antibodies were stored at 4℃” and “The chemical powders were stored at 4℃. They were dissolved in DMSO at 10 mg/ml. The diluted chemicals were stored at 4℃” (lines 347-353, page 10).

  1. There is a need to elaborate on the extraction or isolation procedures of DNA/RNA.

As suggested, we have now added more detail to DNA/RNA extraction as following:

According to the manufacturer's instructions for the FastPure Cell/Tissue DNA isolation Mini Kit (Vazyme, Nanjing, China) or the total RNA rapid extraction kit (Fastagen, Shanghai, China), total DNA and RNA were extracted from cells or culture supernatants as a reported method [10]. In brief, the samples were collected in 2 mL microfuge tubes, lysis solution was added, and the lysates were carefully aspirated into the separation columns and centrifuged. Subsequently, elution buffer was added and centrifuged to elute the total DNA or RNA (lines 368-374, page 10).

  1. Provide information on the normalization method used for quantifying protein expression levels.

As suggested, we have now provided information on the normalization method as: According to the manufacturer's instructions for the BCA protein assay kit, the protein concentrations of the samples are calculated according to the standard curve generated on the day [53] (lines 402-404, page 11).

  1. State the statistical software utilised in the data analysis.

As suggested, we have now stated the statistical software as: using the GraphPad Prism 8 software (line 422, page 11).

Conclusion

Ideally, this section should outline the study's findings and suggest implications for implementation. The conclusion provided in this paper is too short, and the following are areas that could be improved upon for clarity:

  1. Give definite results to confirm the assertion that FAN actually suppresses ASFV replication at micromolar dilutions.
  2. Proffer any likely clinical applications and/ or implications of using FAN as a new approach to treat ASFV infection. Address any challenges that may occur in rendering these results into clinical practice.
  3. State any constraints of the study that may negatively impact the interpretation of the results or the likely usage of FAN as an antiviral agent.

As suggested, we have now modified this Conclusion as: In summary, our study demonstrates that FAN effectively suppresses ASFV replication at micromolar concentrations in PAMs through inhibiting ASFV-induced activation of AKT/mTOR/NF-κB signaling pathway. FAN may potentially be used as a novel agent to treat ASFV infection and inhibiting the AKT/mTOR pathway could be a novel strategy to develop anti-ASFV agents. Given that the in vivo antiviral effect of one compound is closely associated with a variety of factors, including its pharmacokinetic properties, achievable concentration in virus-infected tissues and its tolerated dose, it will be necessary to conduct well designed pre-clinical studies to investigate FAN’s antiviral effect against ASFV infection in swine. (lines 426-434, page 11).

References

Consider presenting references for established methods and protocols mentioned in the paper, particularly if they are not commonly published. This will assist readers in replicating the experiments accurately.

We have modified all references as suggested.

Reviewer 2 Report

Comments and Suggestions for Authors

'Fangchinoline inhibits ASFV replication by suppressing the AKT/mTOR/NF-κB signaling pathway in porcine alveolar macrophages' by Su et al., is a nice paper that studies the role of FAN in the treatment of ASF and finds that this compound can be a promising therapeutic agent due to its capacity to inhibit the AKT/mTOR/ NF-κB pathway. The study is done in isolated porcine macrophage cells. Although the study has its limitations being performed in isolated macrophage system, it is promising foundation work which can be expanded in in vivo animal models. The experiments are carefully done and data is clean and convincing. I would request the authors to add a model figure in the end which would explain their findings into one diagram.

Author Response

Reviewer 2

Reviewer #2: Fangchinoline inhibits ASFV replication by suppressing the AKT/mTOR/NF-κB signaling pathway in porcine alveolar macrophages' by Su et al., is a nice paper that studies the role of FAN in the treatment of ASF and finds that this compound can be a promising therapeutic agent due to its capacity to inhibit the AKT/mTOR/ NF-κB pathway. The study is done in isolated porcine macrophage cells. Although the study has its limitations being performed in isolated macrophage system, it is promising foundation work which can be expanded in in vivo animal models. The experiments are carefully done and data is clean and convincing. I would request the authors to add a model figure in the end which would explain their findings into one diagram.

As suggested, we have now added our working model as Figure 7 (on page 9).

Figure 7. Schematic overview of the inhibitory mechanisms of FAN on ASFV replication. ASFV infection induces the activation of cellular AKT/mTOR/NF-κB signaling pathway, and FAN suppresses AKT phosphorylation and thereby inhibits ASFV replication in PAMs.

Reviewer 3 Report

Comments and Suggestions for Authors

The manuscript by Su et al., describes the antiviral activity of Fangchinoline ( an alkaloid derived from the dried root of a medicinal plant, Stephaniae tetrandrine) against the African swine fever virus. Further, the authors demonstrate that Fangchinoline inhibits the virus-activated AKT/mTOR/NF-κB signaling pathway and not by directly inactivating the viral particles.

The study is well designed and executed with proper controls included in the experiments. Discussion is brief and conclusions are drawn based on the reported findings.

Comments on the Quality of English Language

minor grammar edits required

Author Response

Responses to Reviewers′ Comments

Reviewer 3

Reviewer #3: The manuscript by Su et al., describes the antiviral activity of Fangchinoline ( an alkaloid derived from the dried root of a medicinal plant, Stephaniae tetrandrine) against the African swine fever virus. Further, the authors demonstrate that Fangchinoline inhibits the virus-activated AKT/mTOR/NF-κB signaling pathway and not by directly inactivating the viral particles.

The study is well designed and executed with proper controls included in the experiments. Discussion is brief and conclusions are drawn based on the reported findings.

We thank Reviewer 3 for reviewing this manuscript patiently.

Reviewer 4 Report

Comments and Suggestions for Authors

Fangchinoline inhibits ASFV replication by suppressing the 2 AKT/mTOR/NF-κB signaling pathway in porcine alveolar mac- 3 rophages

Dear author and editor:

The article talked about evaluation of the anti-viral activity of Fangchinoline against ASFV virus. Using an in vitro model. The article could be published after a minor revision.

I have some comments on it:

·      Is there any anti-viral activity of fanchinoline against other viruses, is there a direct effect or other possible mechanism of action? you can discuss this point in the discussion.

·      the author can talk about pathogenesis of the virus in the introduction.

Thank you very much, best regards.

Author Response

Responses to Reviewers′ Comments

Reviewer 4

Reviewer #4: Fangchinoline inhibits ASFV replication by suppressing the 2 AKT/mTOR/NF-κB signaling pathway in porcine alveolar macrophages.

Dear author and editor:

The article talked about evaluation of the anti-viral activity of Fangchinoline against ASFV virus. Using an in vitro model. The article could be published after a minor revision.

I have some comments on it:

Is there any anti-viral activity of fanchinoline against other viruses, is there a direct effect or other possible mechanism of action? you can discuss this point in the discussion. The author can talk about pathogenesis of the virus in the introduction.

In the Introduction, we actually have mentioned antiviral activities of FAN against other viruses as following: In recent years, FAN was found to possess antiviral activities against porcine epidemic diarrhea virus (PEDV) [21], human immunodeficiency virus type 1 (HIV) [22], Middle East respiratory syndrome coronavirus (MERS-CoV) and severe acute respiratory syndrome coronavirus 2 (SARS-CoV-2) [23]. Nevertheless, the molecular mechanisms underlying FAN's action against these viral infections remain poorly understood (lines 96-101, pages 2-3).

As suggested, we have now added lines to introduce the pathogenesis of ASFV infection in the Introduction, it reads as following:

ASFV mainly infects and replicates in swine macrophages. Viral replication relies on a highly coordinated process, including controlled expression of different viral genes and the complex interplay between host cellular pathways and viral proteins. The inflammatory response induced by ASFV plays an important role in the pathogenesis of ASFV. It is reported that ASFV induces macrophages to release TNFα, IL-1β, IL-6, IL-8 and other inflammatory cytokines via activating the NF-κB signaling pathway, and the excessive production of inflammatory factors (called as “cytokine storm”) accounts for the dominant responsibility for the death of ASFV-infected pigs [5]. However, the molecular mechanisms underlying ASFV pathogenesis remain largely unknown. (lines 50-58, page 2).

Round 2

Reviewer 1 Report

Comments and Suggestions for Authors

IJMS-2976305-peer-review-report-v2

The authors have demonstrated commendable progress in incorporating the earlier suggestions. With a few more refinements, this paper is poised to advance to the next stage of review and potentially secure publication. I've outlined additional suggestions below:

a.     Line 21: For correct grammatical usage, change the verb form of ‘causes’ to ‘cause’.

b.     Line 38: for correct article usage, add the article ‘the’ just after by (African swine fever (ASF), caused by the African swine fever …..).

c.      Line 39: The article ‘an’ is redundant in this sentence, so it should be deleted.

d.    Line 49: The word ‘both’ is redundant in this sentence, so it should be deleted.

e.     Lines 52/53: Rephrase this sentence to make it clearer to your readers. Suggestion: The inflammatory response induced by ASFV plays an important role in its pathogenesis.

f.      Line 56: The word ‘as’ should be changed to ‘a’.

g.     Line 65: Change the word ‘is’ to ‘are’.

h.     Lines 159/160: Change the wording ‘didn’t show any’ to ‘showed no’.

i.       Line 201: Change the wording ‘particularly’ to ‘particular’.

j.       Line 305: For correct grammatical usage, change ‘to inhibit’ to ‘of inhibiting’.

Author Response

Responses to Reviewers′ Comments

Reviewer 1

Reviewer #1:

The authors have demonstrated commendable progress in incorporating the earlier suggestions. With a few more refinements, this paper is poised to advance to the next stage of review and potentially secure publication. I've outlined additional suggestions below:

  1. Line 21: For correct grammatical usage, change the verb form of ‘causes’ to ‘cause’.

Modified.

  1. Line 38: for correct article usage, add the article ‘the’ just after by (African swine fever (ASF), caused by the African swine fever …..).

Modified.

  1. Line 39: The article ‘an’ is redundant in this sentence, so it should be deleted.

Modified.

  1. Line 49: The word ‘both’ is redundant in this sentence, so it should be deleted.

Modified.

  1. Lines 52/53: Rephrase this sentence to make it clearer to your readers. Suggestion: The inflammatory response induced by ASFV plays an important role in its pathogenesis.

Modified.

  1. Line 56: The word ‘as’ should be changed to ‘a’.

Modified.

  1. Line 65: Change the word ‘is’ to ‘are’.

Modified.

  1. Lines 159/160: Change the wording ‘didn’t show any’ to ‘showed no’.

Modified.

  1. Line 201: Change the wording ‘particularly’ to ‘particular’.

Modified.

  1. Line 305: For correct grammatical usage, change ‘to inhibit’ to ‘of inhibiting’.

Modified.
